# Patients' attitudes toward artificial intelligence (AI) in cancer care: A scoping review protocol

Daniel Hilbers[1], Navid Nekain[1], Alan T. Bates[2,3], John-Jose Nunez[2,3]*

**1** Faculty of Medicine, University of British Columbia, Vancouver, BC, Canada, **2** Department of Psychiatry, University of British Columbia, Vancouver, BC, Canada, **3** BC Cancer, Vancouver, British Columbia, Canada

* johnjose.nunez@ubc.ca

**Data Availability Statement:** No datasets were generated or analysed during the current study. All relevant data from this study will be made available upon study completion.

## Abstract

### Background

Artificial intelligence broadly refers to computer systems that simulate intelligent behaviour with minimal human intervention. Emphasizing patient-centered care, research has explored patients' perspectives on artificial intelligence in medical care, indicating general acceptance of the technology but also concerns about supervision. However, these views have not been systematically examined from the perspective of patients with cancer, whose opinions may differ given the distinct psychosocial toll of the disease.

### Objectives

This protocol describes a scoping review aimed at summarizing the existing literature on the attitudes of patients with cancer toward the use of artificial intelligence in their medical care. The primary goal is to identify knowledge gaps and highlight opportunities for future research.

### Methods

This scoping review protocol will adhere to the Preferred Reporting Items for Systematic Reviews and Meta-Analyses guidelines (PRISMA-ScR). The electronic databases MEDLINE (OVID), EMBASE, PsycINFO, and CINAHL will be searched for peer-reviewed primary research articles published in academic journals. We will have two independent reviewers screen the articles retrieved from the literature search and select relevant studies based on our inclusion criteria, with a third reviewer resolving any disagreements. We will then compile the data from the included articles into a narrative summary and discuss the implications for clinical practice and future research.

### Discussion

To our knowledge, this will be the first scoping review to map the existing literature on the attitudes of patients with cancer regarding artificial intelligence in their medical care.

**Funding:** The author(s) received no specific funding for this work.

**Competing interests:** John-Jose Nunez has received research funding from an unrestricted research grant from Pfizer Canada. Alan T. Bates reported receiving unrestricted grant funding from Pfizer Inc to BC Cancer allocated to the Psychiatry Department during the conduct of the study. Daniel Hilbers and Navid Nekain have no competing interests to declare. This does not alter our adherence to PLOS ONE policies on sharing data and materials.

## Introduction

Artificial intelligence (AI) is a general term used to describe a computer system modelling intelligent behaviour with minimal human intervention [1]. Prior work has investigated using subfields of AI in medicine to harness various types of data [2]. Applications include generative AI, which can generate new language or other data, and predictive AI, which can predict specific outcomes. AI approaches have been investigated to assist human-model interaction in many areas of medicine including surgical consultation, medical education, disease diagnosis, and pharmaceutical research and treatment [3]. The application of AI in oncology has grown substantially in recent years [4, 5]. Cancer is the second leading cause of death worldwide, and disproportionately affects individuals with restricted healthcare access, especially in rural areas and low- and middle-income countries [6–8]. AI has the potential to improve access to specialized care and address these health inequities, particularly by enhancing early detection and facilitating remote diagnostics [4].

Within cancer care, researchers have used AI to predict survival, for predictive supportive care needs, and to improve detection and diagnosis of cancer [9–12]. Specifically, AI algorithms have demonstrated high accuracy in identifying malignant lung nodules, assisting radiologists in improving early detection and treatment decisions [13]. In colorectal cancer, AI has been employed to detect polyps and other early-stage cancer indicators via colonoscopy and histopathology analysis [14]. In mammography, AI enhances breast cancer detection by improving radiologists' performance and accuracy particularly for early-stage cancers [15].

To ensure these advances deliver patient-centered care, we must consider patients' perspectives on how AI should be used in medical care [16]. Patients with cancer can experience a distinct psychosocial toll, given the extended duration, severity, and chronic nature of their illness [17, 18]. This experience may influence their attitudes on the use of AI in their cancer care that may be different from other medical care they receive. To date, a mixed methods systematic review explored patients from many fields of medicine and public attitudes toward AI [19]. The review found that patients and the public generally accepted the use of AI in their medical care but expressed concerns and preferred human supervision of such models. This review found some studies involving patients with cancer, including those finding that patients had a positive attitude toward AI in melanoma diagnostics in dermatology when AI preserves the human physician-patient relationship [20, 21]. Another study found that Chinese patients with cancer generally accepted the use of AI but preferred the oncologist over the AI when their opinions diverged [22]. This review provided an overview of patients and the public's attitudes on AI in their care and briefly discussed this from the perspective of a patient with cancer, but the attitudes of patients with cancer were not the focus of the review or explored in depth. We also identified a narrative review examining patients' perceptions of AI used in breast cancer diagnosis [23]. The review outlined positive patient sentiments toward AI improving diagnostic accuracy, but patients had concerns regarding the trustworthiness of an AI diagnosis and preferred that AI to complement the radiologist rather than replace them. These reviews only examined a few studies of AI in cancer care, leading to a need for a review to more comprehensively assess patients' attitudes toward AI in cancer care beyond the fields of skin and breast cancer screening.

To address this gap, this protocol outlines a systematic scoping review that will collate prior work to understand the attitudes of patients with cancer toward AI in their care. Specifically, this scoping review will update and narrow the focus of Young and colleagues by including only patients with cancer [19]. Also, this review will expand the focus of Pesapane and colleagues by including patients with all cancer types and explore papers that address patients' perspectives beyond only the use of AI in imaging for breast cancer screening [23].

We utilised a proposed conceptual framework for understanding how patients evaluate AI in healthcare to establish our search criteria [24]. According to this framework, we included the search terms: "experiences", "beliefs", and "attitudes" and added additional relevant search terms to ensure a comprehensive search. Our protocol seeks to establish a systematic scoping review that will be inclusive but rigorous by following widely adopted practices for a systematic scoping review.

## Methods

The objective of this review is to map existing literature, identify knowledge gaps, and outline opportunities for future research. To achieve this, we will conduct a scoping review, which is particularly suited for mapping the breadth of available research and identifying gaps where evidence is limited [25]. Scoping reviews are also useful for clarifying definitions of key concepts in emerging fields, providing an overview of the research landscape, and highlighting areas where further study is needed [26, 27]. Additionally, they can facilitate stakeholder engagement by summarizing existing knowledge and guiding future research directions based on ongoing developments [27, 28]. We registered this protocol with the Open Science Framework (https://osf.io/8zph9).

The scoping review will follow a six-stage methodological framework and will report using the Preferred Reporting Items for Systematic Reviews and Meta-Analyses guideline (PRISMA-ScR) [26, 29].

### Stage 1: Identifying the research question

We utilized a proposed conceptual framework for understanding how patients evaluate AI in healthcare to identify our research question [24]. The framework emphasizes exploring the experiences, beliefs, and attitudes to understand how patients evaluate AI in their care. In addition to these concepts, we included other search terms to ensure a comprehensive search. Our primary research question is "What is known from the existing literature about the attitudes of patients with cancer regarding the use of artificial intelligence in their medical care?"

### Stage 2: Identification of relevant studies

Our eligibility criteria are outlined in Table 1 using the Population-Concept-Context framework. The following electronic databases will be searched: MEDLINE (OVID), EMBASE (OVID), PsycINFO (EBSCOHost), and CINAHL (EBSCOHost).

**Table 1. Eligibility criteria according to Population-Concept-Context framework.**

| | |
|---|---|
| Population | Adult patients with cancer including all forms and stages of cancer. We will include a mixed population of patients with and without cancer, if the patients with cancer population is majority, or if the population is composed of those with cancer and those with non-malignant tumours receiving specialized care. We will include mixed populations of patients with cancer and physicians or caregivers, only if patient attitudes are reported independently. |
| | AI must be included and is defined as a general term used to describe a computer system modelling intelligent behaviour with minimal human intervention [1]. |
| Concept | Attitudes on the use of AI in cancer care. Input is received from patients that can broadly be understood as thoughts, feelings, emotions, perspectives, attitudes, opinions, sentiments, beliefs, and experiences. |
| Context | Clinical setting: All settings including inpatient, outpatient, primary care and specialized. |
| | Geography: No limits |
| | Publication type: Primary qualitative and quantitative research published in peer-reviewed journals. Language: English. Publication date: from inception up to July 22, 2024. |

## Stage 3: Search strategy and study selection

Following Joanna Briggs Institute's (JBI) recommended three step search strategy, we conducted an initial limited search of MEDLINE (OVID) and EMBASE (OVID) [23]. Based on this initial limited search, relevant articles were selected and keywords in the title and abstract, along with relevant index terms were identified. These keywords and index terms were then used to develop a final search strategy and to conduct a literature search across the identified databases for this review. Furthermore, the reference lists of literature included in this scoping review will be searched for additional relevant studies. The final search strategy found in Table 2 was developed with support from subject librarians at the University of British Columbia.

Study Selection: Using Covidence, at least two independent reviewers will screen the titles and abstracts of the articles identified by the literature search. Afterwards, the full text review of the relevant articles will be reviewed by the same reviewers and the reviewers will determine if the article meets our criteria. Our inclusion and exclusion criteria were determined using the Population-Concept-Context (PCC) framework [30]. Disagreements between reviewers will be resolved by consensus or by a third independent reviewer on the team.

## Stage 4: Charting the data

A tabular chart will be developed by two independent reviewers to collect relevant information from the selected articles. Table 3 outlines a preliminary list of PCC [30].

## Stage 5: Collating, summarizing and reporting the results

We will perform Stage 5 according to recommended practice [31].

1. Collating and summarizing the results: We will consolidate the results into a summary and discuss their relevance to our research question.

2. Reporting the results: We will structure the outcomes based on study objectives, methodological approaches, key findings, and identified gaps in the existing literature.

3. Explore implications for future research: This scoping review will guide future research by examining the attitudes of patients with cancer toward the use of AI in their medical care. [25, 26, 32].

We will report the findings in accordance with the Preferred Reporting Items for Systematic Reviews and Meta-Analyses extension for Scoping Reviews checklist.

## Stage 6: Consultation with stakeholders

We will consult with stakeholders and share the preliminary findings with patients with cancer, clinicians, and other relevant parties at an upcoming research summit workshop [26]. Additionally, we plan to host Deliberative Dialogue sessions to facilitate thoughtful and reasoned discussions about AI in cancer care [33]. These sessions will enable diverse partners to engage with and utilize the insights from this scoping review.

Engaging stakeholder consultation will enhance the relevance and applicability of our findings in several key ways [26]. As this is a scoping review, our aim is to map existing literature, identify knowledge gaps, and outline opportunities for future research. Stakeholder input will ensure that the research addresses the most pressing concerns of patients, clinicians, and other relevant groups [33]. By incorporating these perspectives, we aim to highlight potential gaps and additional areas of interest that may not be fully represented in the literature. Additionally,

**Table 2. List of search terms.**

| | MEDLINE (OVID) |
|---|---|
| **Cancer** | (cancer* or neoplas* or malignan* or tumor*).tw,kw,kf. |
| | OR |
| | exp Neoplasms/ |
| | AND |
| **Artificial Intelligence** | (AI OR "artificial intelligence" or "computational intelligence" or "computer reasoning" or "computer vision systems" or ("knowledge Acquisition" adj2 computer*) or ("knowledge representation" adj2 computer*) or "Machine Intelligence" or "machine learning" or "natural language processing" or "computer neural network*" or "computational neural network*" or "connectionist model*" or "models neural network*" or "neural networks*" or perceptron* or "deep learning" or "hierarchical learning" or "data mining" or "text mining").tw,kw,kf. |
| | OR |
| | exp "Artificial Intelligence"/ or exp "Data Mining"/ |
| | AND |
| **Patients' Perspective** | (patient* adj4 ("critical thinking" OR perception* OR perspective* OR emotion* OR feeling* OR regret* OR attitude* OR opinion* OR sentiment* OR "mental process*" OR trust OR distrust OR belief* OR experience*)).tw,kw,kf. |
| | OR |
| | Attitude/ or exp Emotions/ |
| | EMBASE (OVID) |
| **Cancer** | (cancer* or neoplas* or malignan* or tumor*).ab,kf,ti. |
| | OR |
| | exp Neoplasms/ |
| | AND |
| **Artificial Intelligence** | (AI OR "artificial intelligence" or "computational intelligence" or "computer reasoning" or "computer vision systems" or ("knowledge Acquisition" adj2 computer*) or ("knowledge representation" adj2 computer*) or "Machine Intelligence" or "machine learning" or "natural language processing" or computer neural network* or computational neural network* or connectionist model* or models neural network* or neural networks* or perceptron* or "deep learning" or "hierarchical learning" or "data mining" or "text mining").ab,kf,ti. |
| | OR |
| | exp "Artificial Intelligence"/ or exp "Data Mining"/ |
| | AND |
| **Patients' Perspective** | (patient* adj4 ("critical thinking" OR perception* OR perspective* OR emotion* OR feeling* OR regret* OR attitude* OR opinion* OR sentiment* OR "mental process*" OR trust OR distrust OR belief* OR experience*)).ab,kf,ti. |
| | OR |
| | Attitude/ or exp Emotion/ |
| | PsychInfo (EBSCO) |
| **Cancer** | cancer* or neoplas* or malignan* or tumor* |
| | OR |
| | DE "Neoplasms" OR DE "Benign Neoplasms" OR DE "Breast Neoplasms" OR DE "Childhood Neoplasms" OR DE "Digestive System Neoplasms" OR DE "Endocrine Neoplasms" OR DE "Leukemias" OR DE "Lung Neoplasms" OR DE "Metastasis" OR DE "Nervous System Neoplasms" OR DE "Skin Neoplasms" OR DE "Terminal Cancer" |
| | AND |

(*Continued*)

**Table 2.** (Continued)

| MEDLINE (OVID) | |
|---|---|
| **Artificial Intelligence** | "AI" or "artificial intelligence" or "computational intelligence" or "computer reasoning" or "computer vision systems" or ("knowledge Acquisition" N2 computer*) or ("knowledge representation" N2 computer*) or "machine intelligence" or "machine learning" or "natural language processing" or "computer neural network*" or "computational neural network*" or "connectionist model*" or "models neural network*" or "neural networks*" or perceptron* or "deep learning" or "hierarchical learning" or "data mining" or "text mining" |
| | OR |
| | DE "Data Mining" OR DE "Text Analysis" OR DE "Artificial Intelligence" OR DE "Affective Computing" OR DE "Artificial Intelligence Ethics" OR DE "Cognitive Computing" OR DE "Computer Assisted Diagnosis" OR DE "Computer Linguistics" OR DE "Computer Vision" OR DE "Expert Systems" OR DE "Fuzzy Logic" OR DE "Heuristics" OR DE "Intelligent Agents" OR DE "Knowledge Representation" OR DE "Machine Learning" OR DE "Robotics" |
| AND | |
| **Patients' Perspective** | patient* N4 ("critical thinking" or perception* or perspective* or emotion* or feeling* or regret* or attitude* or opinion* or sentiment* or "mental process*" or trust or distrust or belief* or experience*) |
| | OR |
| | DE "Attitudes" OR DE "Emotions" OR DE "Affective Valence" OR DE "Emotional Content" OR DE "Emotional Health" OR DE "Emotional Intelligence" OR DE "Emotional Processing" OR DE "Emotional Regulation" OR DE "Emotional Responses" OR DE "Emotional States" OR DE "Emotional Style" OR DE "Emotional Support" OR DE "Expressed Emotion" |
| CINAHL (EBSCO) | |
| **Cancer** | cancer* or neoplas* or malignan* or tumor* |
| | OR |
| | MH "Neoplasms+" |
| AND | |
| **Artificial Intelligence** | "AI" or "artificial intelligence" or "computational intelligence" or "computer reasoning" or "computer vision systems" or ("knowledge Acquisition" N2 computer*) or ("knowledge representation" N2 computer*) or "Machine Intelligence" or "machine learning" or "natural language processing" or "computer neural network*" or "computational neural network*" or "connectionist model*" or "models neural network*" or "neural networks*" or perceptron* or "deep learning" or "hierarchical learning" or "data mining" or "text mining" |
| | OR |
| | MH "Data Mining+" OR MH "Artificial Intelligence+" |
| **Patients' Perspective** | patient* N4 ("critical thinking" or perception* or perspective* or emotion* or feeling* or regret* or attitude* or opinion* or sentiment* or "mental process*" or trust or distrust or belief* or experience*) |
| | OR |
| | MH "Attitude" OR MH "Emotions+" |

**Table 3. Preliminary table of charting elements and associated questions for data.**

| General information | Population | Context | Concept |
|---|---|---|---|
| Author(s) | Demographics (age, sex, ethnicity, socio-economic) | Community (urban/rural/mixed) | Patients' attitudes toward AI |
| Year of publication | Inclusion and exclusion criteria | Clinical setting (inpatient/outpatient; primary care/specialized) | Type of AI |
| Country of publication | Population and sample size | Country | |
| Publication type | Type of cancer | | |
| Aims/purpose | Cancer stage | | |
| Study design | | | |
| Recruitment procedures | | | |

stakeholders can assist in interpreting our findings by offering context that might be over-looked [34]. This may highlight opportunities for future research, as discussions are likely to uncover gaps and propose areas for further investigation [35]. Specifically, we will seek stakeholder input to help integrate our findings with future efforts to implement AI tools in cancer care.

## Discussion

AI is being investigated for a diversity of applications in cancer care. To ensure we use these tools in a patient-centered manner, it is essential to understand patients' perspectives and attitudes toward AI in their treatment, especially as AI becomes a topic more readily discussed in the public sphere. Initial studies have found that patients both accept the use of AI in medical care but have concerns about it operating without physician supervision [36]. Given the distinct complexity of cancer care, including its impacts on psychosocial health and its combination of chronicity and severity, we believe a focus of this question on cancer care is warranted [37]. For example, patient perspectives may differ if AI tools are being used to influence treatment decisions that can directly impact their chances of surviving cancer, compared to being used to guide treatment decisions for a low-acuity condition. To our knowledge, this will be the first scoping review to map the existing literature on the attitudes of patients with cancer regarding AI in their medical care. We anticipate that this scoping review will reveal gaps in the existing literature and provide direction to better understand patient's needs to guide clinical deployment of AI in cancer care. Based on prior research, we anticipate that cancer patients would generally express optimism about the use of AI in their care. However, we also expect heightened concerns regarding its unsupervised use, potential privacy issues, and specific themes related to end-of-life care, particularly whether AI might be employed to discontinue curative-intent treatments.

Our intention with this review is to include a broad set of studies in adult cancer care. We believe that deferring cancer care for children and adolescents is justified due to the unique characteristics of this population, including the distinct types of cancers they face, expected differences in technological familiarity and comfort, as well as variations in autonomy and their capacity to provide informed consent [38, 39].

Our search criteria are based on a framework proposed for examining patients' attitudes regarding artificial intelligence in healthcare [24]. According to this framework, we included the search terms: "experiences", "beliefs", and "attitudes". To ensure a comprehensive search, we included additional relevant search terms identified after an initial review of terms used in the literature that are similar.

A potential limitation of the novelty of this scoping review may be finding results that are similar to reviews conducted for general medical care. However, given the specific nature of cancer, we believe investigating whether themes are similar or different is warranted. Other potential limitations of this scoping review include excluding papers that are not in English, secondary research, and grey literature. We recognize that findings from secondary research could be relevant, but we chose to exclude these papers to avoid duplication. The exclusion of grey literature was necessary to keep the review practical. Despite these exclusions, we believe our decisions will make the review focused and manageable, contributing to clear and specific conclusions that can better inform future research, policy-making, and clinical practice.

## Supporting information

**S1 File. PRISMA-P 2015 checklist.**
(DOCX)

**S2 File. Preferred Reporting Items for Systematic reviews and Meta-Analyses extension for Scoping Reviews (PRISMA-ScR) checklist.**
(DOCX)

## Acknowledgments

The authors are grateful to the University of British Columbia librarian, Jane Jun, for her contributions to the search strategy.

## Author Contributions

**Conceptualization:** Daniel Hilbers, John-Jose Nunez.

**Methodology:** Daniel Hilbers, Navid Nekain, John-Jose Nunez.

**Project administration:** Daniel Hilbers.

**Visualization:** Daniel Hilbers, John-Jose Nunez.

**Writing – original draft:** Daniel Hilbers.

**Writing – review & editing:** Daniel Hilbers, Navid Nekain, Alan T. Bates, John-Jose Nunez.

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
