## [Decision Letter · Decision Letter 0]

2 Sep 2024

PONE-D-24-34039Patients’ attitudes towards artificial intelligence (AI) in cancer care: a scoping review protocolPLOS ONE

Dear Dr. Nunez,

Thank you for submitting your manuscript to PLOS ONE. After careful consideration, we feel that it has merit but does not fully meet PLOS ONE’s publication criteria as it currently stands. Therefore, we invite you to submit a revised version of the manuscript that addresses the points raised during the review process.

**Please address all reviewer comments, specifically R1 has major concerns on method and results. thanks. **==============================

We look forward to receiving your revised manuscript.

Kind regards,

Daswin De Silva

Academic Editor

PLOS ONE

Journal Requirements:

"I have read the journal's policy and the authors of this manuscript have the following competing interests: John-Jose Nunez has received research funding from an unrestricted research grant from Pfizer Canada. Alan T. Bates reported receiving unrestricted grant funding from Pfizer Inc to BC Cancer allocated to the Psychiatry Department during the conduct of the study. Daniel Hilbers and Navid Nekain have no competing interests to declare."

Reviewers' comments:

Reviewer's Responses to Questions

**Comments to the Author**

1. Does the manuscript provide a valid rationale for the proposed study, with clearly identified and justified research questions?

Reviewer #1: Yes

Reviewer #2: Yes

2. Is the protocol technically sound and planned in a manner that will lead to a meaningful outcome and allow testing the stated hypotheses?

Reviewer #1: Partly

Reviewer #2: Yes

3. Is the methodology feasible and described in sufficient detail to allow the work to be replicable?

Reviewer #1: No

Reviewer #2: Yes

4. Have the authors described where all data underlying the findings will be made available when the study is complete?

Reviewer #1: Yes

Reviewer #2: Yes

5. Is the manuscript presented in an intelligible fashion and written in standard English?

Reviewer #1: Yes

Reviewer #2: Yes

6. Review Comments to the Author

You may also provide optional suggestions and comments to authors that they might find helpful in planning their study.

Reviewer #1: Line 72-73: Refers to specific authors without providing a citation number, which would allow the reader to draw a link between the author and a presumably prior mentioned studies.

Line 75-77: There is likely to be a high degree of overlap between attitudes and perspectives vs emotions and feelings both within the respective combinations and between them. Inclined to disagree that exploration of emotions and feelings is a novel contribution in this context.

Line 89: makes a sweeping comment about multiple areas which benefit from this specific type of review. It is not clear to me how a scoping review in particular can do what is claimed in line 89. requires more clarity.

Methods section :

Stage 5 lacks relevant details on undertaking the review such as whether inclusion and exclusion criteria will be determined using a framework such as PICO. What information will be extracted for quantitative vs qualitative vs mixed methods studies and whether a rating tool for quality of studies such as the MMAT will be used for this purpose.

Stage 6: This step does not add methodological rigor to the research undertaken, as it is about dissemination of the findings, beyond completion of the work.

Line 170: reference is made to end of life care, which is a stage of cancer. if this is part of the focus of the review, table 3 should also extract information regarding the stage of cancer, in addition to cancer type.

Line 174-175: Are these the key distinctions between adult vs child and adolescent demographics? Consider aspects such as ability to provide informed consent here.

While there is likely to be value in the proposed work, the paper will benefit from consideration given to the above, and attention paid to grammar, typos, tense. Further detail is needed in the methods section to increase replicability of this protocol.

Reviewer #2: This review presents opinion analysis of using AI in cancer care from the perspective of the patient. It is an important perspective to consider given the integration of AI in cancer care.

Following are some comments to improve:

1. Consider using terms like privacy/security in search terms as these are major concerns for patients

2. Explain how your study will change based on the consultation with stakeholders. What impact will it have? Clear explanation on this process will be important.

3. Consider including the stage in the cancer journey to the analysis (Context column in Table 3) as it can provide valuable insights, such as use of AI in diagnosis/treatment/post-care etc. Patients' opinions might differ based on for what purpose AI is used for.

7. PLOS authors have the option to publish the peer review history of their article (what does this mean?). If published, this will include your full peer review and any attached files.

Reviewer #1: No

Reviewer #2: No

---

## [Author Response · Author response to Decision Letter 0]

17 Sep 2024

Dear Reviewers and Editor, 

Thank you for your consideration and valuable feedback for our manuscript. 

We have addressed each reviewer comment in our rebuttal letter, and included the relevant changes to the text. 

In particular, we thank Reviewer #1 for their suggestions around adding further methodological details to increase the rigor of this work. We have now added multiple extra details to stages 3, 4, and 6 of the methodology to ensure the methods are well described and replicable. 

With thanks, 

Daniel Hilbers, Navid Nekain, Alan T. Bates, and John-Jose Nunez

---

## [Decision Letter · Decision Letter 1]

22 Oct 2024

PONE-D-24-34039R1Patients’ attitudes towards artificial intelligence (AI) in cancer care: a scoping review protocolPLOS ONE

Dear Dr. Nunez,

Thank you for submitting your manuscript to PLOS ONE. After careful consideration, we feel that it has merit but does not fully meet PLOS ONE’s publication criteria as it currently stands. Therefore, we invite you to submit a revised version of the manuscript that addresses the points raised during the review process.

**Please address the minor revisions noted by R1, thank you for addressing all others from round 1.**==============================

We look forward to receiving your revised manuscript.

Kind regards,

Daswin De Silva

Academic Editor

PLOS ONE

Journal Requirements:

Reviewers' comments:

Reviewer's Responses to Questions

**Comments to the Author**

1. Does the manuscript provide a valid rationale for the proposed study, with clearly identified and justified research questions?

Reviewer #1: Partly

2. Is the protocol technically sound and planned in a manner that will lead to a meaningful outcome and allow testing the stated hypotheses?

Reviewer #1: Partly

3. Is the methodology feasible and described in sufficient detail to allow the work to be replicable?

Reviewer #1: No

4. Have the authors described where all data underlying the findings will be made available when the study is complete?

Reviewer #1: Yes

5. Is the manuscript presented in an intelligible fashion and written in standard English?

Reviewer #1: Yes

6. Review Comments to the Author

You may also provide optional suggestions and comments to authors that they might find helpful in planning their study.

Reviewer #1: The justification for focusing on cancer in this review is likely to benefit from outlining cancer statistics (i.e. mortality and mortality data, inequitable access to care, and the role of AI in addressing inequitable access for people who cannot easily attend specialised centres providing cancer care).

Line 21: I don't agree that "chronicity and severity" is something that differentiates a cancer from "other medical conditions". There is a plethora of other medical conditions which can be equally chronic and severe.

Line 44-45: Diagnosis is not limited to breast cancer. Would be good to expand on the other cancers for which AI is used as well.

Line 143-144: Please reword to make it clear that patients' attitudes are being examined. Currently a grammar error in this sentence.

Stage 6 : Stakeholder consultation as described is not a standard method of adding rigour to a systematic review. A review is conducted to scope the existing literature on a give topic and report on the findings from the literature, alone. Therefore, the statements in line 158 -160 in my opinion are not appropriate in this context, and in fact reduced the credibility of the review, as it is then influenced by external sources of information.

Stakeholder involvement, if at all, should be conducted prior to determining the scope of a review, the research questions and search terms. Not as the final stage.

While it is evident that effort to report more detail on the methodology has been undertaken by the authors, it is important the authors give due consideration to the concerns noted above, as these concerns have a direct impact on the methodology of this protocol as well as the resultant findings reported from this review.

7. PLOS authors have the option to publish the peer review history of their article (what does this mean?). If published, this will include your full peer review and any attached files.

Reviewer #1: No

---

## [Author Response · Author response to Decision Letter 1]

12 Dec 2024

Reviewer #1: 

The justification for focusing on cancer in this review is likely to benefit from outlining cancer statistics (i.e. mortality and mortality data, inequitable access to care, and the role of AI in addressing inequitable access for people who cannot easily attend specialised centres providing cancer care).

Thank you for your feedback. We agree that outlining relevant cancer statistics such as mortality and access to care will greatly benefit the review’s justification for focusing on cancer. We have addressed this by adding text to the manuscript at the following lines in the introduction (Line 45):

“The application of AI in oncology has grown substantially in recent years (4,5). Cancer is the second leading cause of death worldwide, and disproportionately affects individuals with restricted healthcare access, especially in rural areas and low- and middle-income countries (6-8). AI has the potential to improve access to specialized care and address these health inequities, particularly by enhancing early detection and facilitating remote diagnostics (4).”

Line 21: I don't agree that "chronicity and severity" is something that differentiates a cancer from "other medical conditions". There is a plethora of other medical conditions which can be equally chronic and severe.

Thank you for bringing up this great point. We have addressed this by changing the text to:

“However, these views have not been systematically examined from the perspective of patients with cancer, whose opinions may differ given the distinct psychosocial toll of the disease.”

We have also opted to edit line 50 to change the word “unique” to “distinct” as we agree with your point mentioned here and believe that distinct better describes the psychosocial toll patients with cancer face.

“Patients with cancer can experience a distinct unique psychosocial toll, given the extended duration, severity, and chronic nature of their illness (10,11).”

Line 44-45: Diagnosis is not limited to breast cancer. Would be good to expand on the other cancers for which AI is used as well.

We agree that we should expand on other cancers for which AI is used. We have included the following text:

“Specifically, AI algorithms have demonstrated high accuracy in identifying malignant lung nodules, assisting radiologists in improving early detection and treatment decisions (8). In colorectal cancer, AI has been employed to detect polyps and other early-stage cancer indicators via colonoscopy and histopathology analysis (9). In mammography, AI enhances breast cancer detection by improving radiologists’ performance and accuracy particularly for early-stage cancers (10).”

Line 143-144: Please reword to make it clear that patients' attitudes are being examined. Currently a grammar error in this sentence.

Thank you for identifying this. We have adjusted the text to:

“Explore implications for future research: This scoping review will guide future research by examining the attitudes of patients with cancer attitudes toward the use of AI in their medical care (18,19,25).”

Stage 6 : Stakeholder consultation as described is not a standard method of adding rigour to a systematic review. A review is conducted to scope the existing literature on a give topic and report on the findings from the literature, alone. Therefore, the statements in line 158 -160 in my opinion are not appropriate in this context, and in fact reduced the credibility of the review, as it is then influenced by external sources of information.

Stakeholder involvement, if at all, should be conducted prior to determining the scope of a review, the research questions and search terms. Not as the final stage.

Thank you for identifying this and we value your input regarding the reduced credibility of involving stakeholder consultation in a scoping review. We have adjusted our text below to move away from emphasizing that stakeholder consultation adds methodological rigor. Specifically, we emphasize how this will improve the relevance and applicability of our research and perhaps assist in meeting one of our objectives of identifying areas for future research. We would like to emphasize that we are following a gold standard methodological approach for scoping reviews which suggests that stakeholder consultation is essential and should be conducted as stage 6 (21).

“Engaging stakeholders will enhance the relevance and applicability of our findings in several key ways (21). As this is a scoping review, our aim is to map existing literature, identify knowledge gaps, and outline opportunities for future research. Stakeholder input will ensure that the research addresses the most pressing concerns of patients, clinicians, and other relevant groups (28). By incorporating these perspectives, we aim to highlight potential gaps and additional areas of interest that may not be fully represented in the literature. Additionally, engaging stakeholders can assist in interpreting our findings by offering context that might be overlooked (29). This may highlight opportunities for future research, as discussions are likely to uncover gaps and propose areas for further investigation (30). Specifically, we will seek stakeholder input to help integrate our findings with future efforts to implement AI tools in cancer care.”

While it is evident that effort to report more detail on the methodology has been undertaken by the authors, it is important the authors give due consideration to the concerns noted above, as these concerns have a direct impact on the methodology of this protocol as well as the resultant findings reported from this review.

We are grateful for your time in reviewing our paper. You have brought forward some excellent points and we have done our best to adjust our paper accordingly.

---

## [Editor Report · Decision Letter 2]

26 Dec 2024

Patients’ attitudes towards artificial intelligence (AI) in cancer care: a scoping review protocol

PONE-D-24-34039R2

Dear Dr. Nunez,

We’re pleased to inform you that your manuscript has been judged scientifically suitable for publication and will be formally accepted for publication once it meets all outstanding technical requirements.

Kind regards,

Daswin De Silva

Academic Editor

PLOS ONE
---

## [Editor Report · Acceptance letter]

3 Jan 2025

PONE-D-24-34039R2 

PLOS ONE

Dear Dr. Nunez, 

I'm pleased to inform you that your manuscript has been deemed suitable for publication in PLOS ONE. Congratulations! Your manuscript is now being handed over to our production team.

Kind regards, 

on behalf of

Prof. Daswin De Silva 

Academic Editor

PLOS ONE